# Distributionally Robust Imitation Learning

**Mohammad Ali Bashiri**     **Brian D. Ziebart**     **Xinhua Zhang**
Department of Computer Science
University of Illinois at Chicago
Chicago, IL 60607
{mbashi4, zhangx, bziebart}@uic.edu

## Abstract

We consider the imitation learning problem of learning a policy in a Markov Decision Process (MDP) setting where the reward function is not given, but demonstrations from experts are available. Although the goal of imitation learning is to learn a policy that produces behaviors nearly as good as the experts' for a desired task, assumptions of consistent optimality for demonstrated behaviors are often violated in practice. Finding a policy that is distributionally robust against noisy demonstrations based on an adversarial construction potentially solves this problem by avoiding optimistic generalizations of the demonstrated data. This paper studies Distributionally Robust Imitation Learning (DRoIL) and establishes a close connection between DRoIL and Maximum Entropy Inverse Reinforcement Learning. We show that DRoIL can be seen as a framework that maximizes a generalized concept of entropy. We develop a novel approach to transform the objective function into a convex optimization problem over a polynomial number of variables for a class of loss functions that are additive over state and action spaces. Our approach lets us optimize both stationary and non-stationary policies and, unlike prevalent previous methods, it does not require repeatedly solving an inner reinforcement learning problem. We experimentally show the significant benefits of DRoIL's new optimization method on synthetic data and a highway driving environment.

## 1   Introduction

We consider the imitation learning setting of learning to perform a task based only on demonstrations that are provided by experts. There are two main approaches often considered for this learning problem: *Behavioral Cloning* [19] and *Inverse Reinforcement Learning* [18, 1, 24, 14]. In behavioral cloning, the learner attempts to learn a policy in a supervised learning manner, in which a direct mapping from states to actions is estimated from the demonstrated trajectories. Behavioral cloning, while simple, often generalize poorly when attempting to predict goal-directed sequential decisions due to compounding errors caused by covariate shift and only tends to succeed when given large amounts of data [22, 23]. Alternatively, inverse reinforcement learning (IRL) rationalizes demonstrated trajectories by estimating a reward function that makes the expert's policy optimal. The problem of determining the reward function is inherently ill-posed, since a single policy can be optimal for multiple reward functions.

To obtain a unique solution in IRL, many methods have been proposed, with the maximum entropy principle [31, 32] and margin maximization [20] being two widely employed methods among researchers. Maximum (causal) entropy IRL [32], in particular, seeks an entropy-maximizing distribution over sequences that matches expected feature counts [1] with those observed from demonstrations. For imitation learning in very high dimensional and continuous spaces, where function approximations such as deep neural networks are often used, IRL methods have generally

35th Conference on Neural Information Processing Systems (NeurIPS 2021).

been less efficient than behavioral cloning methods [13] since they require reinforcement learning as an inner loop. Recent adversarial IRL methods [10, 9, 25], however, seem more likely to be effective.

A common assumption in imitation learning is that all expert behaviors have the same level of trustworthiness and are optimal/near-optimal [30]. However, it is common for noisy expert behaviors to violate this optimality assumption in practice. Thus, relying heavily on the optimality of the expert's behavior may degrade an imitation learner, making it prone to failure [30]. Existing methods have proposed to inject noise into the expert's policy demonstrations to obtain a more robust policy [15], or to train a discriminator that distinguishes between expert trajectories and generated trajectories that learns reward functions that are robust to changes in dynamics [11].

An alternative approach to obtain a robust policy, which we adopt in this work, is to search for a policy that is distributionally robust given the training data. For this purpose, the learner's policy is obtained by solving a game between a learner and an adversary [5], where the learner seeks to minimize a loss defined between them, and the adversary seeks to maximize this loss by choosing a distribution over policies subject to a set of constraints that match statistics from the training data. This approach leverages two uncertainty sets as opposed to typical Distributionally Robust Optimization (DRO) methods [4, 6] where the uncertainty set is only defined over the adversary (demonstrator-estimator) and the learner's policy is assumed to have a specific parametric form. Previous work [5] proposed to solve imitation learning under an adversarial formulation using the Double Oracle method [17] to solve the corresponding optimization. However, this method may take up to exponential time in the size of state-action space and decision horizon.

In this paper, we connect Distributionally Robust Imitation Learning (DRoIL) and Maximum Entropy Inverse Reinforcement Learning (MaxEnt IRL) by showing that MaxEnt IRL is a special case of DRoIL when a certain loss function and policy description is used. We then show that DRoIL is a framework for maximizing a *general* entropy function that is defined based on a particular loss of interest. We then cast DRoIL's objective function into a convex optimization problem over a polynomial number of variables, which is simpler to understand and implement and also significantly improves the training time. We extend the formulation to stationary policies, which enables us to experimentally show the benefits of learning a robust policy in a highway driving simulation.

## 2   Preliminaries

We model sequential decision making problems using discrete Markov Decision Processes (MDPs). A MDP $\mathcal{M}$ is specified by a tuple $(\mathcal{S}, \mathcal{A}, \Gamma, \mathcal{R}, \gamma, P_0)$ where: $\mathcal{S}$ and $\mathcal{A}$ are state and action (control input) spaces (assumed to be finite, $|\mathcal{S}|, |\mathcal{A}| < \infty$); $P_0$ is the initial state distribution; $\Gamma$ represents the transition probabilities, the state distribution upon taking action $a$ in state $s$, $P(s'|a, s)$; reward function $\mathcal{R} : \mathcal{S} \times \mathcal{A} \rightarrow \mathbb{R}$; and discount factor $\gamma \in (0, 1]$. We assume feature vectors $\phi : \mathcal{S} \times \mathcal{A} \rightarrow [0, 1]^d$ over state-action pairs that capture the most salient properties distinguishing preferred and nonpreferred trajectories where $\mathcal{R}$ can be written as a (linear) function of these feature vectors given a reward vector $\mathbf{w}$: $\mathcal{R}(s, a) = \mathbf{w} \cdot \phi(s, a)$.

A policy $\pi \in \Pi$ is the probability of taking action $a$ in state $s$, $\pi(a, s) = P(a|s)$ and $\Pi$ represents the set of all possible stochastic policies. Demonstrations by an expert are given as a set of trajectories $\mathcal{D} = \{\tau^1, \cdots, \tau^m\}$. A trajectory is a sequence of state-action pairs $\tau = (s_0, a_0, \cdots, s_T, a_T)$ over horizon $T$. From an optimization perspective, *Imitation Learning with a General Loss*, finds a policy $\hat{\pi}$ that minimizes the difference between learner behavior and expert behavior $\pi_E$:

$$\hat{\pi} \in \operatorname*{argmin}_{\pi} \mathcal{L}(\pi, \pi_E),^1$$

where $\mathcal{L}$ measures the dissimilarity between two policies' behaviors.

Having access to only sample demonstrations of a policy for training, it is essential to quantify the behavior of a policy. Commonly used in behavioral cloning, one approach is to measure the marginal distribution of states and actions, $P(s, a)$ induced by a policy. Note that in the case of infinite-horizon MDPs, expected (discounted) number of visits to state-action pairs is used. A more desirable measure for long horizons is the expectation of trajectory features [1]. This approach is often used in inverse reinforcement learning, where a reward function of the features is learned. The expected (discounted

---
[1]We overload $\mathcal{L}$ for any type of loss in different contexts (e.g., loss between deterministic policies or states).

if $\gamma \in (0, 1))$ feature of a policy $\pi$ (similar to [1]) is defined as:

$$\mu(\pi) = \mathbb{E}\left[\sum_{t=0}^{T}\gamma^t\phi(s_t, a_t)\big|\pi, \Gamma\right] \in \mathbb{R}^d. \tag{1}$$

## 3 Robust Imitation Learning and Maximum Entropy

Foundational work [27, 12] has developed a close relationship between robust Bayes decisions, maximum entropy, and minimizing worst-case expected loss also known as adversarial learning. Several works have followed this framework of robustness for different supervised learning problems, such as cost-sensitive classification [2], multivariate loss prediction [29], ordinal regression [7], and graphical models [8]. For the imitation learning setting, [5] has applied the adversarial learning framework on inverse reinforcement learning problem for a restricted class of loss functions and proposed a double oracle algorithm [17] to solve the corresponding optimization problem.

In the following, we define a general framework of distributional robustness for imitation learning to obtain the policy that robustly minimizes an imitative loss:

**Definition 1.** *Given an imitative loss function $\mathcal{L}$ that measures the distance of two policies' behavior,* **Distributionally Robust Imitation Learning (DRoIL)** *is defined as a two-player zero-sum game between the learner and the demonstrator, in which each player chooses a distribution over control policies - constructing a stochastic policy, $\hat{\pi}$ and $\check{\pi} \in \tilde{\Xi}$, then the players receive the loss between the behaviors $\mathcal{L}(\hat{\pi}, \check{\pi})$ as their payoff. The minimax strategy for the learner is given by:*

$$\min_{\hat{\pi}} \max_{\check{\pi} \in \tilde{\Xi}} \mathcal{L}(\hat{\pi}, \check{\pi}), \tag{2}$$

*where $\tilde{\Xi}$ is a set of constraints characterized by the demonstrated data.*

Generally, $\tilde{\Xi}$ can be in the form of moment matching in (1) that is commonly used in inverse reinforcement learning:

$$\check{\pi} \in \tilde{\Xi} \leftrightarrow \mathbb{E}\big[\sum_{t=0}^{T}\gamma^t\phi(s_t, a_t)\big|\check{\pi}, \Gamma\big] = \tilde{\mu} \triangleq \mathbb{E}\big[\sum_{t=0}^{T}\gamma^t\phi(s_t, a_t)\big|\pi_E, \Gamma\big],$$

where $\pi_E$ represents the policy that demonstrated trajectories are generated from.

This formulation assumes that except for certain properties of the limited samples of available demonstrated behavior, the demonstrator's policy is the worst-case possible for the learner. This approach avoids generalizing from available demonstrations in an optimistic manner that may be unrealistic and lead to a policy that does not work well in practice, especially in situations where there is noise in the demonstrated behaviors.

The minimax formulation in Definition 1 is closely related to the principle of maximum entropy, which is used in Maximum Entropy Inverse Reinforcement Learning (MaxEnt IRL) [31]. MaxEnt IRL provides a probabilistic approach under the constraint of matching the reward value of demonstrated behavior to resolve the ambiguity in choosing a distribution over decisions. Under this model, trajectories with equivalent rewards have equal probabilities, and trajectories with higher rewards are exponentially more preferred according to the following:

$$P(\zeta_i|\mathbf{w}) = \frac{1}{Z(\mathbf{w})}e^{\mathbf{w}^\top\phi_{\zeta_i}}, \tag{3}$$

where $\zeta_i$ and $\phi_{\zeta_i}$ represent a trajectory and its corresponding sum of features.

Note that a distribution over trajectories (paths) is an alternative policy description that provides a stochastic policy where the probability of an action is weighted by the expected exponentiated rewards of all paths that begin with that action:

$$P(a|\mathbf{w}) \propto \sum_{\zeta:a\in\zeta_{t=0}} P(\zeta|\mathbf{w}).$$

To show the connection between DROIL and MaxEnt IRL, we employ a tool from the general theory of exponential families [3] that shows for certain classes of two-player zero-sum games, there exists a parametric distribution for the minimax strategy, as shown in Lemma 1.

**Lemma 1** (Barndorff-Nielsen [3]). *Let $p \in \Xi$ be a probability distribution over space $X$, where: $\Xi = \{p : \mathbb{E}_p(\mathcal{T}(X)) = C\}$ describes a mean-value constraint; $\mathcal{T}(X)$ represents a vector value statistic; and $C$ is a constant. Let $q$ be also a distribution chosen from the set of all probability mass functions. For the maximum entropy distribution, $\max_{q \in \tilde{\Xi}} \min_p \mathbb{E}[-\log q(X)]$, $p^* = q^*$ exists as a parameterized function of the form $p^* = \exp\{\alpha_0 + \boldsymbol{\alpha}^\top \mathcal{T}(X)\}$ with parameters $\alpha_0$ and $\boldsymbol{\alpha}$.*

Equipped with Lemma 1, we develop a connection between DROIL and MaxEnt IRL in Theorem 1.

**Theorem 1.** *The stochastic policy $P(\zeta_i|\mathbf{w})$ obtained from MaxEnt IRL in Equation (3) is obtained from DROIL minimax strategy in Definition 1 when the logarithmic loss is used.*

*Proof.* We construct a stochastic policy in Definition 1 as a probability distribution over all possible trajectories $\{\zeta_1, ..., \zeta_M\}$. Let $\hat{\pi} = P(\zeta)$ and $\check{\pi} = Q(\zeta)$ be the probability distributions of the learner and the demonstrator, respectively. The expected feature of demonstrator now can be written as $\mu_{\check{\pi}} = \sum_i Q(\zeta_i)\phi_{\zeta_i}$, which resembles the mean value constraint in Lemma 1, therefore, for logarithmic loss, $P^*(\zeta_i) = Q^*(\zeta_i) = \exp\{w_0 + \mathbf{w}^\top \phi_{\zeta_i}\}$. $\qquad\qquad\square$

We can generalize beyond the logarithmic loss and resulting maximum entropy policy (Theorem 1) to other losses of interest by using Generalized Entropy functions, $H(P) := \inf_Q \mathbb{E}[\mathcal{L}(P, Q)]$. Proposition 1 describes the relation between DROIL and generalized maximum entropy.

**Proposition 1.** *For any policy descriptions and loss functions that DROIL in Definition 1 has Nash equilibrium, $\hat{\pi}$ is a robust action and $\check{\pi}$ is the maximizer of a **generalized** entropy function.*

Proposition 1 provides a general approach to resolve the ambiguity of matching constraints—where many policies lead to the same feature counts—by choosing either a policy $\check{\pi}$ that does not exhibit any additional preferences beyond matching feature expectations with respect to a loss of interest $\mathcal{L}$ (maximum generalized entropy) or a policy $\hat{\pi}$ that minimizes the worst-case expected loss. This can be seen as Maximum Generalized Entropy Inverse Reinforcement Learning, where the choice of loss function is not restricted to the logarithmic loss function.

We extend the following lemma that describes a class of loss functions and policies for which the solution of the game in Definition 1 exists, thus Proposition 1 can be applied.

**Lemma 2.** *For the game in Definition 1, where learner $\hat{\pi}$ and demonstrator $\check{\pi}$ simultaneously choose a stochastic policy, if the loss function is additive and the payoff can be written as a bilinear function: $\sum_{i=1}^n \sum_{j=1}^m l_{ij}\hat{\pi}_i\check{\pi}_j$, the game has a solution and minimax (maximin) strategy producing the solution as long as $m$ and $n$ are finite.*

*Proof.* In his generalized minmax theorem, Von Neumann [28] shows that bilinear games with arbitrary nonempty closed, bounded convex sets of actions have saddle points as long as $m$ and $n$ are finite. Assuming the demonstrated data are generated from a policy (optimality not necessary) the convex moment matching constrained in Definition 1 keeps $\tilde{\Xi}$ closed, bounded, and non-empty. $\quad\square$

**Example 1.** *One illustrating example is to let each player's pure strategy to be a deterministic policy $\delta$ and define the loss as the expectation over a given loss between each individual deterministic policies. A mixed strategy now represents a stochastic policy $\pi$, which is a probability distribution over the set of all deterministic policies and Equation (2) can be written as a bilinear game with payoff $\sum_{i=1}^n \sum_{j=1}^m l_{ij}P(\hat{\delta}_i)P(\check{\delta}_j)$ where $l_{ij}$ is $\mathcal{L}(\hat{\delta}_i\check{\delta}_j)$.*

Lemma 2 describes a restricted class of loss functions (e.g., that the logarithmic loss does not belong to) for which the resulting policy is the maximizer of a more flexible concept of entropy. A class of loss functions of interest are loss functions that are additive over state and action spaces. Similar to Example 1, previous work [5] constructs a policy using a distribution over deterministic policies and for additive loss function by writing Equation (2) as a bilinear game and solving the corresponding optimization. We also focus on this class of loss functions. However, we provide an efficient optimization algorithm for this class of loss functions that avoids the exponential worst-case time complexity of the previous double oracle [17] approach.

## 4 Learning and Inference

To solve the optimization problem in Equation (2), one needs to specify the policy description $\pi$, and the distance measure between two policies' behaviors $\mathcal{L}$. The choice of $\pi$ and $\mathcal{L}$ can result in different algorithmic approaches.

### 4.1 Double Oracle

In [5], the authors construct the learner and the demonstrator stochastic policies as mixtures of deterministic non-stationary policies. Concretely, let $\Upsilon = \{\delta_1, \delta_2, ...\}$, be the set of all possible non-stationary deterministic policies, then $\hat{\pi}_\Delta := P(\{\hat{\delta} \in \Upsilon\})$ and $\check{\pi}_\Delta = P(\{\check{\delta} \in \Upsilon\})$. Note that $|\Upsilon| \in \mathcal{O}(|\mathcal{A}|^{|\mathcal{S}|T})$ which is exponential in size of state space and time horizon, $|\mathcal{S}|$, $T$. Assuming the loss function is additive over state-action pairs, [5] shows that Equation (2) can be written as a bilinear game:

|  | $\check{\delta}_1$ | $\check{\delta}_2$ | $\cdots$ | $\check{\delta}_j$ |
|---|---|---|---|---|
| $\hat{\delta}_1$ | $\mathcal{L}(\hat{\delta}_1, \check{\delta}_1)$ $+\psi(\check{\delta}_1)$ | $\mathcal{L}(\hat{\delta}_1, \check{\delta}_2)$ $+\psi(\check{\delta}_2)$ | $\cdots$ | $\mathcal{L}(\hat{\delta}_1, \check{\delta}_j)$ $+\psi(\check{\delta}_j)$ |
| $\hat{\delta}_2$ | $\mathcal{L}(\hat{\delta}_2, \check{\delta}_1)$ $+\psi(\check{\delta}_1)$ | $\mathcal{L}(\hat{\delta}_2, \check{\delta}_2)$ $+\psi(\check{\delta}_2)$ | $\cdots$ | $\mathcal{L}(\hat{\delta}_2, \check{\delta}_j)$ $+\psi(\check{\delta}_j)$ |
| $\vdots$ | $\vdots$ | $\vdots$ | $\ddots$ | $\vdots$ |
| $\hat{\delta}_i$ | $\mathcal{L}(\hat{\delta}_i, \check{\delta}_1)$ $+\psi(\check{\delta}_1)$ | $\mathcal{L}(\hat{\delta}_i, \check{\delta}_2)$ $\psi(\check{\delta}_2)$ | $\cdots$ | $\mathcal{L}(\hat{\delta}_i, \check{\delta}_j)$ $\psi(\check{\delta}_j)$ |

Table 1: Payoff matrix of $G(\hat{\pi}_\Delta, \check{\pi}_\Delta | \mathbf{w})$ with loss function $\mathcal{L}$, deterministic policies $\delta$ and Lagrangian potentials $\psi$.

$$\min_{\mathbf{w}} \min_{\hat{\pi}_\Delta} \max_{\check{\pi}_\Delta} \overbrace{\sum_i \sum_j p(\delta_i)p(\delta_j)\mathcal{L}(\hat{\delta}_i, \check{\delta}_j) + \sum_j p(\delta_j) \underbrace{\mathbf{w} \cdot \mathbb{E}[\phi|\check{\delta}_j]}_{\psi(\check{\delta}_j)}}^{G(\hat{\pi}_\Delta, \check{\pi}_\Delta | \mathbf{w})} - w \cdot \tilde{\boldsymbol{\mu}}.$$

This will result in a matrix game of exponential size in the number of states $\mathcal{S}$ and time horizon, as shown in Table 1. To obtain the Lagrange variables $\mathbf{w}$, this matrix game needs to be repeatedly solved to compute the gradient with respect to $\mathbf{w}$. This requires solving a linear program with $\mathcal{O}(|\mathcal{A}|^{|\mathcal{S}|T})$ variables with a simplex constraint, which is impractical for even modestly sized problems. To mitigate this problem, they employed the double oracle method [17] in an attempt to construct a smaller sub-portion of the matrix by gradually adding pure actions through solving a time-varying control problem. However, there is no guarantee that the support set of Nash-equilibrium of the defined game is small and the algorithm may need to solve up to an exponential number of time-varying optimal control problems.

### 4.2 State-Action Distribution

We propose an alternative way to transform the optimization in Equation (2) into a convex problem. Our approach is based on using state-action marginals to construct the stochastic policies of learner and the demonstrator, where the number of required variables is linear in $|\mathcal{S}|$ and $|\mathcal{A}|$, and it can also be extended to stationary policies. In the following, we first look at the non-stationary case and then extend our method to stationary policies.

A policy $\pi$ induces a probability distribution $P_t(s)$ over the states of an MDP $\mathcal{M}$ at each time step. State-action marginals are similarly defined as $P_t(s, a) = P_t(s)\pi_t(a|s)$. A *valid state-action marginal* is a set of simplices corresponding to each time-step that satisfy the Bellman flow constraints for a given MDP $\mathcal{M}$. Let $P_t(s, a) \in \Delta$ be the probability of state-action pairs $(a, b)$ at time $t$; a valid marginal distribution $\mathbf{P} \in \Omega$ satisfies the following affine constraints: for all $s'$, $\sum_{s,a} P_t(s, a)P(s'|s, a) = \sum_a P_{t+1}(s', a)$, where $P(s'|s, a) \in \Gamma$ and $\Omega$ represents the set of all valid marginal distribution for a given MDP $\mathcal{M}$.

Employing state-action marginals allows us to write the objective function in Equation (2) as a convex problem of $\mathcal{O}(|\mathcal{A}||\mathcal{S}|T)$ variables as shown in Theorem 2 in a vectorized form.

**Theorem 2.** *For an additive loss function over states and actions $\mathcal{L}$, Solving* DROIL *optimization in Equation* (2) *is equivalent to solving the following convex minimax problem over **marginal state-action probabilities** of the learner $\mathbf{P} = (\mathbf{p}_1, ..., \mathbf{p}_T)$ and demonstrator $\mathbf{Q} = (\mathbf{q}_1, ..., \mathbf{q}_T)$*

*parameterized with Lagrange multipliers* $\mathbf{w}$:

$$\min_{\mathbf{w}} \max_{\mathbf{Q} \in \Omega} \min_{\mathbf{P} \in \Omega} \left[ \sum_{t=0}^{T} \mathbf{p}_t^{\top} \mathcal{L} \mathbf{q}_t + \mathbf{w}^{\top} \Phi^{\top} \mathbf{q}_t \right] - \mathbf{w}^{\top} \tilde{\boldsymbol{\mu}}, \tag{4}$$

where $\mathbf{p}_t$ (similarly $\mathbf{q}_t$) is a vector of size $|\mathcal{A}||\mathcal{S}|$ storing marginal probability of state-action pairs at time $t$: $P_t(s,a)$; $\mathcal{L}$ is the general loss function that is defined over state-action pairs $\mathcal{L} : |\mathcal{A}||\mathcal{S}| \times |\mathcal{A}||\mathcal{S}|$. $\phi$ is a $d \times |\mathcal{A}||\mathcal{S}|$ matrix storing the feature function for each state-action pair. We denote system dynamics with $\Gamma$, which is a matrix of size $|\mathcal{A}||\mathcal{S}| \times |\mathcal{S}|$ storing transition probabilities $P(s'|s,a)$.

Intuitively, each player in the above formulation searches over a *valid* state-action distribution to reach an equilibrium. One of the benefits of constructing a policy using marginals is that the feature expectation can be written as an inner product of the state-action distribution and the state-action feature vector. Thus, the demonstrator's expected feature is $\mu(\mathbf{Q}) = \sum_{t=0}^{T} \Phi \mathbf{q}_t$, and the feature matching constraint is realized by minimizing $\mathbf{w}^{\top} (\sum_{t=0}^{T} \Phi \mathbf{q}_t - \tilde{\boldsymbol{\mu}})$ over dual variables $\mathbf{w}$. Writing the objective function in Equation (2) in terms of state-action marginal probabilities reduces the number of variables needed to represent the equilibrium from $\mathcal{O}(|\mathcal{A}|^{|\mathcal{S}|T})$ to $\mathcal{O}(|\mathcal{A}||\mathcal{S}|T)$. The unconstrained optimization over dual variables $\mathbf{w}$ can be solved using any gradient descent method where the gradient is given by $\sum_{t=0}^{T} \Phi \mathbf{q}_t^* - \tilde{\boldsymbol{\mu}}$ and $\mathbf{Q}^*$ is the solution of the following game for the current $\mathbf{w}_t$:

$$G(\mathbf{w}_t) = \max_{\mathbf{Q} \in \Omega} \min_{\mathbf{P} \in \Omega} \left[ \sum_{t=0}^{T} \mathbf{p}_t^{\top} \mathcal{L} \mathbf{q}_t + \mathbf{w}_t^{\top} \Phi^{\top} \mathbf{q}_t \right]. \tag{5}$$

## 4.3 Stationary Policy

Our approach extends with some modifications to the stationary policy setting. Stationary policies are desirable because they are simpler to describe, and are more natural and intuitive in terms of the behavior that they prescribe. Similar to state-action marginals, we utilize an *occupancy measure* $\rho_\pi : \mathcal{S} \times \mathcal{A} \to \mathbb{R}^+ \cup 0$ to characterize a stationary policy $\pi$. It is defined as the expected (discounted) number of visits to state-action pair $(s, a)$, when following policy $\pi$ and can be written as a feasible set of affine constraints:

$$\mathcal{G} = \{\rho : \rho \ge 0 \mid \sum_a \rho(s', a) = p_0(s') + \sum_{s,a} \gamma P(s'|s,a)\rho(s,a) \; \forall s \in \mathcal{S}\}, \tag{6}$$

where $P(s'|s,a) \in \Gamma$ and $p_0$ is the distribution of starting states.

For a given additive loss $\mathcal{L}$, with the use of an occupancy measure, we write the expected loss $\mathbb{E}[\mathcal{L}(\pi_1, \pi_2)]$ as $\rho_{\pi_1}^{\top} \mathcal{L} \rho_{\pi_2}$ and the expected discounted feature as $\mu_\pi = \Phi^{\top} \rho_\pi$. Theorem 3 shows how we can write Equation (2) as a convex optimization using occupancy measure:

**Theorem 3.** *For an additive loss function over state and actions $\mathcal{L}$, solving DROIL optimization in Equation (2) is equivalent to solving the following convex minimax problem over **occupancy measures** of the learner $\mathcal{P}$ and demonstrator $\mathcal{Q}$:*

$$\min_{\mathbf{w}} \max_{\mathcal{Q} \in \mathcal{G}} \min_{\mathcal{P} \in \mathcal{G}} \mathcal{P}^{\top} \mathcal{L} \mathcal{Q} + \mathbf{w}^{\top} (\Phi^{\top} \mathcal{Q} - \tilde{\boldsymbol{\mu}}). \tag{7}$$

The convex optimization in Equation (7) can also be solved using any gradient-based method where the gradient is obtained by solving a constrained game with $\mathcal{O}(|\mathcal{A}||\mathcal{S}|)$ variables.

## 4.4 Inferred Policy

In the non-stationary case, after obtaining $\mathbf{w}^*$, one can use either $\mathbf{Q}^*$ or $\mathbf{P}^*$ as the produced non-stationary Markovian stochastic policy by computing $\pi^*(a|s_t) = p_t(a|s) = \frac{p_t(a,s)}{\sum_a p_t(a,s)}$. $\mathbf{Q}$ corresponds to the policy that maximizes the generalized entropy and $\mathbf{P}$ corresponds to the policy that has minimized the worst-case expected loss.

For the stationary case, [26] has proved that there is a one-to-one mapping between $\mathcal{G}$ and $\Pi$, in a sense that for an occupancy measure $\mathcal{P} \in \mathcal{G}$, $\pi(a|s) \triangleq \frac{\mathcal{P}(a,s)}{\sum_a \mathcal{P}(a,s)}$ is the only policy that results in $\mathcal{P}_\pi$. Therefore, one can similarly use $\mathcal{P}$ or $\mathcal{Q}$ as the produced stationary stochastic policy.

For either case, assuming the reward function can be written as $\mathcal{R}(s, a) = \mathbf{w} \cdot \phi(s, a)$, then $\mathbf{w}^*$ plays the role of the reward weight vector that rationalizes the demonstrated behaviors. Consequently, we can use $\mathbf{w}^{*\top} \Phi$ to obtain the optimal reward function $\mathcal{R}^*$ and use an MDP solver to obtain the (deterministic) corresponding policy.

# 5 Optimization

In both stationary and non-stationary cases, any gradient descent method can be used to optimize over dual variables $\mathbf{w}$. To compute the gradient, one needs to compute $\mathbf{Q}^*$ for non-stationary and $\mathcal{Q}^*$. However, since $\mathbf{w}$ is unconstrained, by adding a norm of $\mathbf{w}$ with hyperparameter $\lambda$, one can directly solve $\mathbf{w}$ and replace it in the objective. Since the optimization algorithms are similar in both cases, we only mention the non-stationary case:

$$\min_{\mathbf{w}} \max_{\mathbf{Q} \in \Omega} \min_{\mathbf{P} \in \Omega} \left[ \sum_{t=0}^{T} \mathbf{p}_t^\top \mathcal{L} \mathbf{q}_t + \mathbf{w}^\top \Phi^\top \mathbf{q}_t \right] - \mathbf{w}^\top \tilde{\boldsymbol{\mu}} + \frac{\lambda}{2} \|\mathbf{w}\|^2, \tag{8}$$

and setting: $\mathbf{w} = \frac{\tilde{\boldsymbol{\mu}} - \Phi \sum_{t=0}^{T} \mathbf{q}_t}{\lambda}$ we have:

$$\max_{\mathbf{Q} \in \Omega} -\frac{1}{2\lambda} \left\| \tilde{\boldsymbol{\mu}} - \Phi \sum_{t=0}^{T} \mathbf{q}_t \right\|^2 + \min_{\mathbf{P} \in \Omega} \sum_{t=0}^{T} \mathbf{p}_t^\top \mathcal{L} \mathbf{q}_t, \tag{9}$$

which is a constrained quadratic optimization in $\mathbf{Q}$. Using Danskin's theorem, the gradient of $\mathbf{q}_t$ is given by

$$\frac{1}{\lambda} (\Phi^\top \tilde{\boldsymbol{\mu}} - \Phi^\top \Phi \sum_{t=0}^{1} T \mathbf{q}_t) - \mathcal{L} \mathbf{p}_t^*,$$

where $\mathbf{p}^*$ can be found using linear programming (linear objective with affine constraints) efficiently using standard linear programming toolbox. An alternative approach is to solve the dual of optimization over $\mathbf{p}_t$ and maximize it along with $\mathbf{Q}$.

---

**Algorithm 1** Distributionally Robust Imitation Learning (DRoIL)

---

**Input:** $\mathcal{D} = \{\tau^1, \tau^2, \cdots, \tau^m\}, \Gamma, p_0$
Initialize $\mathbf{Q}^0$, compute $\tilde{\boldsymbol{\mu}}$ using $\mathcal{D}$, and set $i = 0$
**repeat**
    Compute $\nabla_i f(\mathbf{Q}^i)$ in Equation (9)
    $i = i + 1$
    Using $\nabla_i f(\mathbf{Q}^i)$ calculate with $\bar{\mathbf{Q}}^{i+1}$
    **if** $\bar{\mathbf{Q}}^{i+1} \in \Omega$ **then**
        $\mathbf{Q}^{i+1} = \bar{\mathbf{Q}}^{i+1}$
    **else**
        $\mathbf{Q}^{i+1} =$project $(\bar{\mathbf{Q}}^{i+1})$ where projection function is defined in Equation (10)
    **end if**
**until** convergence

---

**Projection Step**

At each iteration of the algorithm, we need to project $\mathbf{Q}$ to a convex domain to maintain a valid state-action distribution given $\Gamma$ and the initial state distribution. Essentially,

$$\min_{\mathbf{Q}} \frac{1}{2} \sum_{t=0}^{T} \|\mathbf{q}_t^* - \mathbf{q}_t\|^2 \quad \text{s.t. } \mathbf{Q} \in \Omega, \tag{10}$$

from which, by using a Lagrangian method and strong duality, we obtain:

$$\max_{\mathbf{V}} \min_{\mathbf{Q} \in \Delta} \sum_{t=0}^{T} \frac{1}{2} \|\mathbf{q}_t^* - \mathbf{q}_t\|^2 + \left( \mathbf{q}_{t-1}^\top \Gamma - \mathbf{q}_t^\top \mathbf{Z} \right) \mathbf{v}_t, \tag{11}$$

where $\mathbf{Z}$ operator computes the state distribution: $\mathbf{q}_t^\top \mathbf{Z} = \sum_a q_t(s, a)$. To compute the gradient $\mathbf{V}$, a quadratic program over $\mathbf{Q}$ with probability simplex constraints needs to be solved. This can be analytically determined by sorting each $\mathbf{q}_t$, which takes $\mathcal{O}(|\mathcal{S}||\mathcal{A}| \log(|\mathcal{S}||\mathcal{A}|))$ time.

# 6 Experimental Results

In our experiments, we compare DRoIL with prior methods on several imitation learning tasks. We investigate: 1) How our convex optimization improves the training time compared to the double oracle method; 2) How the choice of loss function affects the performance of DRoIL; and 3) How accurately DRoIL predicts actions compared to other IRL methods.

## 6.1 Training Time

To compare the training time of our proposed convex optimization with the double oracle approach [5], we adopt their experimental setup in GridWorld. In this experiment, trajectories are collected from simulated navigation across a discrete 2D grid where the agent starts from a random starting point, and navigates through the grid to reach a specified target location. Taking a step in the grid has a cost and the agent's goal is to reach the target location while minimizing the accumulated navigation cost (maximizing the reward). This problem can

| Size | DRoIL Time | DRoIL Cost | DO Time | Do Cost |
|---|---|---|---|---|
| 128 | 1.6 | $-7.39$ | 30.1 | $-7.39$ |
| 432 | 1.8 | $-21.61$ | 42.1 | $-21.74$ |
| 1024 | 6.0 | $-20.0$ | 141.1 | $-20.0$ |
| 2000 | 98.6 | $-30.1$ | 608.1 | $-30.1$ |
| 3500 | 496 | $-39.2$ | 2020 | $-39.1$ |
| 5500 | 881 | $-58.7$ | 4970 | $-58.9$ |

Table 2: Elapsed time in second until convergence with $10^{-3}$ tolerance.

be formulated as an optimal sequential decision-making problem in a finite Markov decision process where the optimal policy is non-stationary. We consider linear cost function $C(s) = \mathbf{w}^{*\top}\phi(s) + \epsilon(s)$, where feature function $\phi(s, a)$ and weight vector $\mathbf{w}^*$ are drawn from $U(0, 1)^d$, and $\epsilon \sim U(0, 1)$. Transition function is non-deterministic with parameter $p_m \in (0, 1]$ which navigates the agent to randomly choose neighbors with probability $(1 - p_m)$.

The loss function for this experiment is set to $\mathbb{E}\left[\sum_{t=0}^{T} \sqrt{(\hat{\mathbf{X}}_t - \check{\mathbf{X}}_t)^2 + (\hat{\mathbf{Y}}_t - \check{\mathbf{Y}}_t)^2}\right]$, where $(X, Y)$ represents the grid position of the agent. We generate trajectories from the optimal policy that is obtained by solving the true reward function and train DRoIL and DO until convergence. Along with expected loss, we report the elapsed time that it takes to converge for our proposed method and double oracle (DO) in Table 2, averaged over eight repetitions of the experiment for this result and our later results. As Table 2 shows, our proposed method requires significantly less training time for the same performance and scales very well with the size of state space.

## 6.2 Loss Function Effect

For the second question, we show that different choices of loss function result in different performance for the learned policy in the imitation learning setting. Therefore, the choice of loss function provides an extra tool to incorporate certain domain knowledge, and design a problem-specific loss that potentially results in better produced policies. For the purpose of comparison of different losses, we revisit the GridWord environment, however, we train several stationary policies with different losses. We consider 0-1 loss, which equally penalizes any mismatch between state-action pairs; action-loss, which incurs loss only when an action differs from another pair in the same state; random loss, which is drawn from a uniform distribution; and finally Euclidean distance between two positions in the grid. It is clear from Figure 1 that the policy produced from Euclidean loss outperform other policies from other losses, which shows the benefit of using a task-specific loss function.

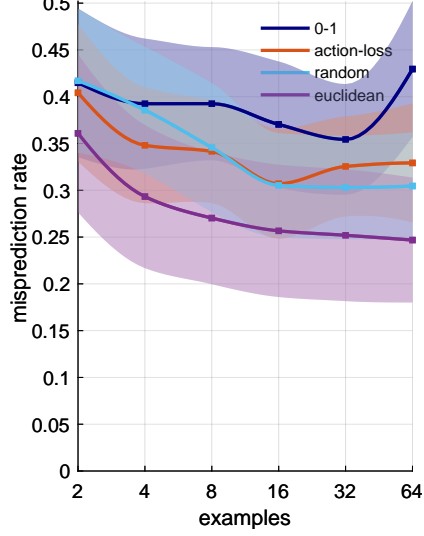

Figure 1: Performance of DRoIL when different loss functions are used.

## 6.3 Highway Driving

To evaluate DROIL in a complex environment with more realistic behavior, we compared it with several other IRL methods in a highway driving simulator with non-linear reward function: MMP [20], the projection algorithm of Abbeel and Ng [1], and LEARCH [21].

Following the setting in Levine et al. [16], the task is to drive a car on a three-lane highway in which the agent can switch lanes and drive at up to four times the speed of traffic while all other vehicles move at a constant speed. The set of features includes the distance to the nearest vehicle in each lane (in front and behind), current speed, and current lane. We also evaluate each method on the original environment and on four additional random environments, denoted as "transfer". We set a uniformly random loss for DROIL and train all methods using examples sampled from the stochastic MaxEnt IRL policy which can intuitively be viewed as noisy samples of an underlying optimal policy. To evaluate the performance of each method, we use the misprediction rate, which is defined as the ratio of incorrect actions compared to the optimal policy, and the expected value difference, which measures the suboptimality of the learned policy under the true reward. Since DROIL is able to produce stochastic policy, with the same argument from [16], we could evaluate the optimal stochastic policies. However, this would unfairly penalize other methods. Therefor, we first obtain the reward weight vector and find the optimal deterministic policy the corresponding reward function. Then, we measure its expected sum of discounted rewards under the true reward function, and subtract this quantity from the expected sum of discounted rewards of the optimal policy.

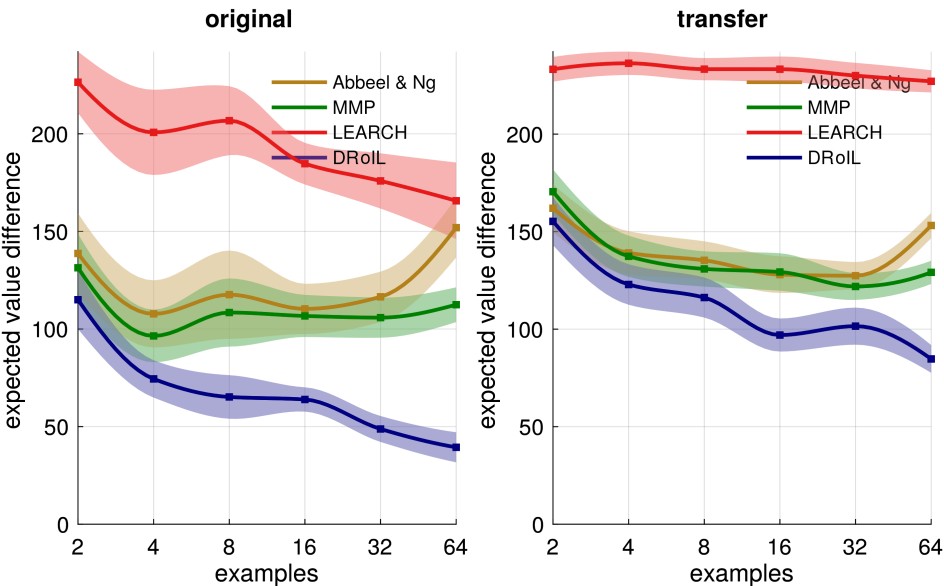

Figure 2: Expected value differences for 64-car-length highways with varying example counts. Lower values are better.

As shown in Figure 2 and Figure 3, DROIL performs very well in terms of the obtained reward and the accuracy of the produced policy in both the original and transfer environments. In contrast, we find that MMP and Abbeel & Ng's approach degrade as the number of examples increase. This matches theory since the suboptimality of the demonstrations becomes more apparent as the number of examples increases. This indicates that under noisy demonstrations (samples from a stochastic policy), a robust approach has the potential to outperform alternative approaches.

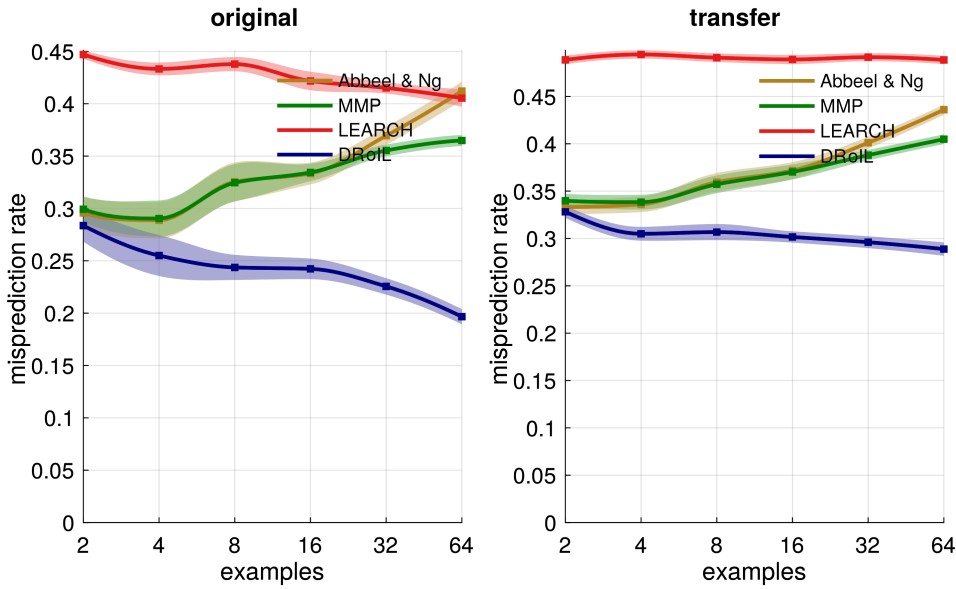

Figure 3: Misprediction rate results for 64-car-length highways with varying example counts. Lower values are better.

## 7    Discussion & Conclusion

We demonstrated a connection between DRoIL, which accepts any loss of interest, and Maximum Entropy Inverse Reinforcement Learning—one of the widely used IRL approaches—which robustly minimizes logarithmic loss. We showed that MaxEnt is a special case of DRoIL framework when logarithmic loss function is used and showed that DRoIL can be seen as the maximizer of a generalized concept of entropy. We provided a novel approach to cast DRoIL's objective into a convex optimization over a polynomial number of variables and experimentally showed our proposed algorithm provides faster training time. DRoIL is naturally designed to perform robustly against noisy demonstrations. Our experiment in the highway driving task showed that when demonstrations are noisy, it robustly learns an appropriate policy.

Improvements in imitation learning have the potential for both societal benefits and harms. For example, better imitating top surgeons could scale their abilities to a broader populations that are medically under-served. Enabling robots that better imitate effective soldiers could cause great harm if used inappropriately. Like all general purpose tools, avoiding intentional harms while still providing benefits is an unsolved challenge. We take the position that providing methods that are more robust to noise will help to avoid *unintentional* harms—the application of methods in a well-intentioned manner that fail to maximize their benefits and may instead produce harms through their fragility.

Our presented experiments are restricted to discrete/low-dimensional decision processes. For future work, we are interested in finding a connection between DRoIL and the Generalized Exponential family and applying DRoIL on very high dimensional state and action spaces that need function approximators such as deep neural networks.

## Acknowledgements

This material is based upon work supported by the National Science Foundation under Grant Nos. 1652530 and 1910146.

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

# A  Proof of Theorem 2 & 3

Suppose the loss function $\mathcal{L}$ is decomposable over state action pairs, then we can write, Equation (2) as following:

$$\min_{\hat{\pi}} \max_{\check{\pi}} \sum_{t=1}^{T} \left[ \sum_{\substack{\hat{s},\hat{a} \\ \check{s},\check{a}}} \left[ P(\hat{S}_t = \hat{s}, \hat{A}_t = \hat{a} | \hat{\pi}, \Gamma) \mathcal{L}(\hat{s}, \check{s}, \hat{a}, \check{a}) P(\check{S}_t = \check{s}, \check{A}_t = \check{a} | \check{\pi}, \Gamma) \right] \right] \tag{12}$$

$$\text{subject to: } \sum_{t=1}^{T} \sum_{\check{s},\check{a}} P_t(\check{S}_t = \check{s}, \check{A}_t = \check{a} | \check{\pi}, \Gamma) \phi(\check{s}, \check{a}) = \tilde{\boldsymbol{\mu}}.$$

By introducing dual variables $\mathbf{w}$ for the feature expectation constraints, the Lagrangian function of Equation (12) is given by:

$$\min_{\hat{\pi}} \max_{\check{\pi}} \min_{\mathbf{w}} \sum_{t=1}^{T} \left[ \sum_{\substack{\hat{s},\hat{a} \\ \check{s},\check{a}}} \left[ P(\hat{S}_t = \hat{s}, \hat{A}_t = \hat{a} | \hat{\pi}, \Gamma) \mathcal{L}(\hat{s}, \check{s}, \hat{a}, \check{a}) P(\check{S}_t = \check{s}, \check{A}_t = \check{a} | \check{\pi}, \Gamma) \right] \right] \tag{13}$$

$$+ \mathbf{w} \cdot \left( \sum_{t=1}^{T} \left[ \sum_{\check{s},\check{a}} P_t(\check{S}_t = \check{s}, \check{A}_t = \check{a} | \check{\pi}, \Gamma) \phi(\check{s}, \check{a}) \right] - \tilde{\boldsymbol{\mu}} \right)$$

The optimization in Equation (13) is over $\hat{\pi}$ and $\check{\pi}$. However, the objective function Equation (13), decomposes over the state-action distribution induced by policies $\hat{\pi}$ and $\check{\pi}$. We directly optimize over the marginals:

$$\min_{(p_1,p_2,...,p_T)} \max_{(q_1,q_2,...,q_T)} \min_{\mathbf{w}} \sum_{t=1}^{T} \left[ \sum_{\substack{\hat{s},\hat{a} \\ \check{s},\check{a}}} \left[ p_t(\hat{s}, \hat{a}) \mathcal{L}(\hat{s}, \check{s}, \hat{a}, \check{a}) q_t(\hat{s}, \hat{a}) \right] \right] \tag{14}$$

$$+ \mathbf{w} \cdot \left( \sum_{t=1}^{T} \left[ \sum_{\check{s},\check{a}} q_t(\check{s}, \check{a}) \phi(\check{s}, \check{a}) \right] - \tilde{\boldsymbol{\mu}} \right),$$

where $p_t(\hat{s}, \hat{a}) = P(\hat{S}_t = \hat{s}, \hat{A}_t = \hat{a})$ and $q_t(\check{s}, \check{a}) = P(\check{S}_t = \check{s}, \check{A}_t = \check{a})$. This optimization needs to be over valid state-action marginals (marginals induced by a policy). So the following constrained need to be satisfied:

$$\Omega := \sum_{\hat{s},\hat{a}} p_t(\hat{s}, \hat{a}) P(\hat{s}'|\hat{s}, \hat{a}) = \sum_{\hat{a}} p_t(\hat{s}', \hat{a}) \quad \forall \quad t, s'$$

$$\text{Similarly for} \quad q_t, \quad \sum_{\check{s},\check{a}} q_t(\check{s}, \check{a}) P(\check{s}'|\check{s}, \check{a}) = \sum_{\check{a}} q_t(\check{s}', \check{a}) \quad \forall \quad t, s'$$

Since Equation (14) is convex in all variables $p_t$, $q_t$, and $\mathbf{w}$, the order of optimization can be changed:

$$\min_{\mathbf{w}} \max_{\mathbf{Q} \in \Omega} \min_{\mathbf{P} \in \Omega} \left[ \sum_{t=1}^{T} \left[ \sum_{\substack{\hat{s},\hat{a} \\ \check{s},\check{a}}} p_t(\hat{s}, \hat{a}) \mathcal{L}(\hat{s}, \check{s}, \hat{a}, \check{a}) q_t(\hat{s}, \hat{a}) + \mathbf{w} \cdot \sum_{\check{s},\check{a}} q_t(\hat{s}, \hat{a}) \phi(\check{s}, \check{a}) \right] \right] - \mathbf{w} \cdot \tilde{\boldsymbol{\mu}},$$

where $\mathbf{P} = (p_1, p_2, ..., p_T)$ and $\mathbf{Q} = (q_1, q_2, ..., q_T)$. $\qquad \square$

The proof for stationary case is similar.

