# OpenReview forum: "Distributionally Robust Imitation Learning"
_NeurIPS.cc/2021/Conference — NeurIPS 2021 Poster_

### Official Review · Reviewer_4tjM · 2021-07-12

**Rating:** 4
**Confidence:** 3

**Summary:**

The paper considers distributionally robust imitation learning (DRIL) which aims to find an imitating policy that is robust against changes in distribution of the expert policy within a feature expectation constraint. Solving this problem gives an imitating policy that minimizes the worst-case performance given a fixed feature map and a loss function. The paper proposes a computationally efficient algorithm to solve DRIL where the main idea is to optimize for state-action distributions instead of policies. The algorithm is empirically evaluated on synthetic and highway driving tasks.

Contributions :

1. A theoretical analysis that gives two insights: 1) MaxEnt-IRL (Zeibart et al., 2008) solves the DRIL problem with the logarithm loss. 2) The solution of DRIL is guaranteed to exist for a class of bilinear loss functions.

2. A computationally efficient algorithm that solves DRIL for MDPs with discrete state-action spaces. The algorithm improves upon the existing algorithm (Chen et al., 2016) in terms of computational efficiency by solving for state-action distributions instead of policies.


**Limitations And Societal Impact:**

Yes.

**Main Review:**

Distributionally robust learning is a promising approach for robust imitation learning. This paper presents a novel theoretical result and a novel algorithm for this. The writing is quite clear and the contents are well structured, but there are issues regarding mathematical clarity. The algorithm is derived using a standard optimization technique, i.e., projection gradient, which leads to a simple and computationally efficient algorithm.

However, the significance of the paper is quite low due to two major issues: 1) lack of comparison against MaxEnt-IRL, and 2) practical limitations of the algorithm. I recommend rejection.

1) The paper firstly claims that MaxEnt-IRL is equivalent to performing DRIL with a logarithmic loss. Then, the paper proposes the algorithm to perform DRIL with different losses. However, it does not explain issues of using logarithm loss for DRIL, i.e., issues of MaxEnt-IRL. Moreover, the proposed algorithm is not empirically compared against MaxEnt-IRL.

2) The algorithm is applicable only to problems with a discrete state-action space. In addition, it also requires a transition probability matrix $\Gamma$ and a fixed feature map $\phi$. These are severe practical limitations when compared to recent researches on imitation learning that are applicable to both discrete and continuous state-action space and does not require a transition probability matrix and a fixed feature map, e.g., AIRL (Fu et al., 2017) which performs model-free MaxEnt-IRL with deep networks.

Minor comments:

- The expectation notation $\mathbb{E}[\cdot]$ is ambigious in the paper since the corresponding distribution is not defined. In particular, I do not understand the expectation in equation 1: $\mathbb{E} [ \mathcal{L} (\hat{\pi}, \check{\pi}) ]$. What is the random variable here and what is the distribution of the random variable for this expectation? I also find it strange that a similar objective in lines 78-79 does not have an expectation.

- The solution $w$ in line 251 is missing a denominator $\lambda$.

- Algorithm 1 refers to $\nabla_i f(Q^i)$ in equation 8, but $f(Q^i)$ is not defined anywhere.

- The experiments are conducted with two simple tasks. I suggest including a large-scale task to make the paper more practically convincing.

- Besides study the effect of loss function, it would be beneficial to also study the effect of feature map. That is, how does changing the feature map change the robustness. In addition, it would be interesting to study the effect of changing the constraint set $\Xi$. For example, $\Xi$ could be a norm-ball around the expert policy.

## Update after rebuttal

I read the rebuttal and other reviews. I thank the authors for the clarification in the response. However, the response does not change my opinion and I still rate the paper as a rejection.

I understand that the paper focus on a particular setting of model-based distribution robust IL in discrete MDPs. However, I think this setting is too limited given the current research frontier in IRL/IL. In addition, evaluation of MaxEnt-IRL should be in the scope of this work since the paper relates MaxEnt-IRL to DRIL.





**Time Spent Reviewing:**

Around 4 hours

---

> ### Author Response · Authors · 2021-08-10
> **Response to questions and concerns**
>
> We thank you for your careful reading and detailed review, we address your questions and concerns in the following:
>
> MaxEnt: MaxEnt is a special case of DRIL. However, from an algorithmic perspective, we only consider loss functions that are additive over states and actions. On the other hand, MaxEnt uses logarithmic loss which is not an additive loss function. For a fair comparison, one might have to choose a task specific loss function and compares DRIL to MaxEnt to show the potential benefits of DRIL. However, since we focus on training time/feasibility and theoretical perspective of DRIL, it seems out of the scope of our work. However, we totally agree our work will benefit from adding such an experiment.
>
> Expectation: We consider a general definition that potentially can deal with: 1. Distribution over policies 2. The stochastic behavior of a policy (e.g. state distribution of a stochastic policy). In line 78-79, we consider a general loss function (overloading) that can measure the distance between to policies’ behavior (so it includes the expectation).
>
>
> Model-Free/Function approximation: As we mentioned, this is one of the early works that tries to use distributional robustness on the imitation learning problem. However, model free IRL and function approximation for continuous/high dimensional state/action spaces is our future direction.
>
> Experimental suggestions: We thank you for your suggestions on new experiments. We take into consideration to apply these ideas and improve our experimental results.

---

### Official Review · Reviewer_KGfa · 2021-07-16

**Rating:** 5
**Confidence:** 2

**Summary:**

This paper considers the problem of robust imitation learning when demonstrations are noisy. The authors propose an adversarial imitation learning optimization problem that results in improved training time compared to a prior double oracle approach. Results show that DRIL outperforms several older imitation learni

**Limitations And Societal Impact:**

yes

**Main Review:**

How is Theorem 2 related to Syed et al. [25]? They also use the state-action occupancies to construct the bellman flow constraints to create a more efficient convex optimization problem. What is different about the current approach?

The paper only compares runtime against the Double Oracle method. It would also be good to compare against a method such as [25] which seems very similar as it also does not require an MDP solver in the inner loop.

Furthermore, the later results only compare against MMP, Abbeel and Ng, and LEARCH. These are all very old algorithms and it's fine to compare against them but they are not considered state of the art. Better baselines including maximum entropy IRL, LPAL [25], and BROIL (Brown et al. 2020) mentioned below would strengthen the results.

Lines 363-369 do not seem correct. They state that the suboptimality becomes more apparent with more demos. However, the demos are from a stochastic max ent policy so optimal actions will always be taken more frequently than suboptimal ones so more data will be good since it will allow the truly optimal actions to be sampled more.

The paper states that DRIL performs robustly to noisy demonstrations. But it is not compared with Max Ent IRL or Bayesian IRL (Ramachandran et al. 2007) both of which are known to be robust to some noise in the demonstrations. These baselines should be added. Another missing related work/baseline is Brown et al. "Bayesian robust optimization for imitation learning." NeurIPS 2020.
It is very similar in motivation as it seeks an imitation learning policy that is distributionally robust. It would be good to compare as a baseline and contrast with the current approach.

Clarity:

Lines 108-110: I don't know what is mean here. Especially the "optimistic manner" part. A little more intuition would be helpful.

Line 339: what is meant by a uniformly random loss? How do you achieve good performance with a random loss function?

Figures 3 have legends that are behind the actual lines making them hard to read.

Typos:

Line 264 *tasks
Table 2 *seconds
Line 345 *Therefore

---
Post Rebuttal:
Thank you for the author response. I think this is an interesting area of work. Unfortunately, I do not think this paper is ready for NeurIPS. Adding more baselines and better comparing to previous works mentioned above is needed. Also adding a more complex domain would significantly strengthen the paper.

**Time Spent Reviewing:**

3

---

> ### Author Response · Authors · 2021-08-10
> **Respond to Review KGfa**
>
> We thank you for your careful reading and detailed review, we address your questions and concerns in the following:
>
> Theorem 1 and 2: Syed et al [25] uses marginal distribution to represent a policy that maximizes the returned value of a policy, assuming the demonstrations are optimal. However, in our approach, we use marginal distributions to represent a policy that minimizes the loss between the learner policy and the worst case estimation of the demonstrated data.
>
> Experimental results: We agree that comparison with more baseline in runtime/policy performance will benefit our work which can be included.
>
> Suboptimal Demonstrations: It is true that more optimal decisions are apparent when the size of demonstrations increase. However, with the same ratio more sub-optimal decisions are included in those data. The experiments show that other algorithms become confused and unable to detect the true decision in such situations.
>
> Other robust baselines: Our focus in this paper was more on the theoretical side of DRIL as well as proposing a feasible training approach. However, we agree adding more baselines will strengthen our paper.
>
> Uniformly random loss: The loss matrix entries are chosen randomly from a uniform distribution. The performance of random loss means that the loss is not task-specific, although it still minimize some notion of distance between the two policies (although it is not task specific).

---

### Official Review · Reviewer_4wnP · 2021-07-19

**Rating:** 5
**Confidence:** 3

**Summary:**

This paper studied distributionally robust imitation learning, and connects the formulation with maximum entropy inverse reinforcement learning.  Based on the objective, the authors transform the optimization problem into a convex optimization problem and it can be efficiently solved by using standar linear programming toolbox. Empirical results show the performance gain using the new formulation compared with prior works.

**Ethical Concerns:**

There is no ethical concerns.

**Ethics Review Area:**

["I don’t know"]

**Limitations And Societal Impact:**

My major concern is the scalability and empirical evaluation of the proposed method. From the paper we can see that the proposed method can not be scalable to complex function approximators, such as neural networks. Besides, if we can many samples (millions of samples) for learning, how will the method perform?
Further, the evaluation domain is too simple, there is even no simple mujoco evaluation experiments.

**Main Review:**

The paper studied distributionally robust imitation learning, where the learners and the demonstrators play a zero-sum game, thus the new learned policy can be robust to distribution shift and noisy demonstrations.

Originality: The formulation is new to me, and indeed the original idea of distributionally robust optimization can solve the distribution shift problem, and can obtain a model that are robust to noisy samples, which would be a good direction to explore for imitation learning.

Quality and clarity: The paper is well written, and the content is well organized, from motivation to problem formulation, and then to the optimization solutions. Finally empirical results show the performance gain of the proposed paper.

Significance: The paper introduces a new perspective for robust imitation learning. while there is a major concern about the scalability of the proposed method, which may limit its general usage in general imitation learning application scenarios.


**Time Spent Reviewing:**

3

---

> ### Author Response · Authors · 2021-08-10
> **Respond to Reviewer 4wnP**
>
> We thank you for your careful reading and review, we address your questions and concerns in the following:
>
> Scalability/Function approximation: As we mentioned, this is one of the early works that tries to use distributional robustness on the imitation learning problem. However, large scale imitation learning using function approximation on continuous/high dimensional state/action spaces is our future direction.

---

### Official Review · Reviewer_rEEu · 2021-08-02

**Rating:** 6
**Confidence:** 3

**Summary:**

This work aims to address the problem of imitation learning where the expert demonstrations may not be optimal and instead near-optimal and somewhat noisy. This problem is approached through the perspective of distributionally robust optimization. This work extends the work of Chen et al. 2016, by reformulating the optimization problem in a manner that results in a significantly more efficient optimization process.

**Ethical Concerns:**

I do not have any immediate concerns.

**Limitations And Societal Impact:**

I believe any negative impacts would be due to the application domain of the method and not inherently due to the work presented here. I do not have any immediate concerns.

**Main Review:**

**Originality:**

This work operate under the formulation of Chen et al. 2016, so the problem formulation is not the novelty of this work. The key novelty of this work is in how the optimization problem is reformulated in order to obtain a more efficient optimization process, which is an important contribution.

**Quality:**

Soundness: Methodology and claims seem valid and sound.

Claims Supported:
- The improvements in runtime are clearly demonstrated in section 6.1
- The performance of DRIL when compared to baseline methods is presented in section 6.3.

Major Comments:
- Lines 363-369: I am not sure I understand why exactly your method is doing better here? Your method is still trying to match the features of the sub-optimal policy, so what in particular about your approach is leading to the improvements? It would be very valuable to include a discussion.
- In section 4.4, you discuss how you can obtain 3 distinct policies from the optimization process. Q: max-ent, P: min worst-case, and lastly a deterministic policy obtained by optimizing the MDP whose reward is given by $\mathcal{R}(s,a) = w^*\cdot\phi(s,a)$. First question: does $w$ define the reward for which Q is optimal, or P is optimal? Second comment: It would be nice if you could provide intuitions for when to use which one (between Q and P and deterministic from $w$). It would be great also if there were experiments that could verify your intuitions.
- It would be nice to also include the max-ent and stochastic worst-case results (policies from Q and P) in section 6.3 as well.

**Clarity:**

This work is generally well-written.

Major Comments:
- Line 339: I'm not sure I understand how you can train your method using a uniform random loss in DRIL? Doesn't the loss need to convey some meaningful notion of similarity?
- Line 338: Please describe what the transfer domains are more accurately instead of "random environments".
- Line 344: Could elaborate 1-2 sentences in addition to saying "with the same argument as [16]".

More Minor Comments:
- Lines 108-111: I'm not sure I understand what this sentence is trying to say "This approach avoids....". Could you clarify? Do you mean that it avoids generalizing (which requires optimism), and this pessimism is unrealistic and leads to a policy that does not work well in practice? Also, can you provide support/evidence for this claim?
- In section 6.3, in the description of the highway environment, it would be useful to also describe how the states and actions are discretized. It would be useful for a reader such as myself not familiar with the MDP.
- Lemma 1: Could you clarify the notation? You have not defined T and C. Also, is the order of max and min not the reverse of the order in equation 1?
- Example 1: What is a pure strategy?

**Significance:**

I think the reformulation of the optimization problem of Chen et al. 2016 in the current form is a significant contribution. First, due to the significant speed-up, and second because this work's formulation appears to be more amenable to eventual application with continuous spaces and neural network function approximators.

Comments:
- Even though the method is not concerned with continuous spaces and function approximators, as a discussion section I think it would be interesting to motivate why this work would be valuable to those settings. Namely, if we manage to transfer this solution to neural networks, would we be able to obtain a method that is importantly different from current adversarial inverse RL literatures (GAIL [Ho et al.], AIRL [Fu et al.], state-marginal matching [Ghasemipour et al.], etc.)?
- One question I have is about author's intentions with section 6.2? I would like to make sure that I am not misunderstanding something. The ability to use differing loss functions does not appear be unique to your methodology. It seems that the same loss functions could have been used by Chen et al. 2016 as well? Am I understanding correctly or is the ability to use varying losses due to the contributions of your work?


**Time Spent Reviewing:**

12

---

> ### Author Response · Authors · 2021-08-10
> **Response to Reviewer rEEU**
>
> We thank you for your careful reading and detailed review, we address your questions and concerns in the following:
>
> Performance in presence of sub-optimal demonstrations: The main advantage of robust learning is that it minimizes a loss with respect to the worst case estimation of the demonstrator behaviors. The demonstrator policy has to match the feature expectation, however, the learner player only minimizes the loss between its policy and the demonstrator. The equilibrium here results in a policy (among all the policies that match the feature expectation) that assumes the demonstrations are not necessarily optimal.
>
> Section 4.4 : $w^*$ shows the weight at the equilibrium when both P and Q are both optimal. However, having $Q^*$ is enough to compute $w^*$. We are not sure exactly when $Q^*$ or $P^*$ is preferred over one another. However, our preliminary experiments show that in some cases $P^*$ performs better than $Q^*$ while in most cases $Q^*$ is the policy with better performance.
>
> Max-Ent and stochastic results: We also think that will be beneficial to our work and make the distinction between them more clear.
>
> Line 339, uniform random loss: The performance of random loss means that the loss is not task-specific, although it still minimizes some notion of distance between the two policies’ behavior. We think it is better to use a non task-specific loss for a fair comparison, although other losses could be added to the experiments.
>
> Transfer Environments: Transfer environments are defined in Levine et al [16]. However, we will add a description to the highway experiment section for more clarity.
>
> Line 334: The argument is that evaluating stochastic policies will unfairly penalize margin-based methods. This can also be added for more clarity.
>
> Line 108: Avoid generalizing means that avoid only mimicking the given demonstrations as much as possible. Within the space of all policies that produce the same feature expectation, the equilibrium of the minimizer of the loss and maximizer of the same loss will result in robust policy that does not only generalize the demonstrated behavior.
>
> Pure strategy: A strategy of a player is a probability over all possible actions. The player play actions randomly according to the probability of each action. A pure strategy is a strategy that only one action (among all possible actions) is played, in other words, a strategy that put probability 1 on a single action.
>
> Discussion on continuous state-action spaces: We believe there is a connection between GAIL and our work when our framework is applied to continuous state-action spaces with function approximation that can be further studied.
>
> Section 6.2: That is right. The ability to take a loss function is not specific to our method or our approach. It can be also used in Chen et al. 2016. Our goal was to show that first our approach has the potential to accept a loss function. Second, we wanted to emphasize that while most of the imitation learning methods do not accept loss function of interest including MaxEnt, the choice of loss function has a large impact on the performance of the imitation learner.

---

### Decision · Program_Chairs · 2021-09-28

**Decision:**

Accept (Poster)

**Comment:**

The rebuttal did not overcome the reviewers' objections. Notably, they are still concerned about the scalability of the proposed approach and the unconvincing experiments (notably the lack of comparison to maxEntIRL, given its relation to the proposed approach). Additional feedback is provided in some reviews (last section of the review).

There is a consensus that this paper is not ready for publication at Neurips, I therefore recommend rejection.

**Consistency Experiment:**

NeurIPS has a long history of experimentation. In 2014, NeurIPS ran an experiment in which 10% of submissions were reviewed by two independent committees to quantify the randomness in the review process. This year, we repeated a variant of this experiment to see how the quality of the review process has changed over time.  This paper was part of the experiment and was therefore assigned to two committees (consisting of reviewers, an Area Chair, and a Senior Area Chair) that reached independent decisions.  If both committees made the same recommendation, this recommendation was followed. If a single committee recommended acceptance, the paper was accepted (with the exception of a few cases in which the other committee identified what we considered a fatal flaw, e.g., an error in a key result).

This copy’s committee reached the following decision: **Reject**

The other committee assigned to the paper recommended **Accept (Poster)**.  You can find the other set of reviews, along with any follow up discussion with the authors here:
https://openreview.net/forum?id=PJEPtZmw-SQ